# Dynamic Compressive and Tensile Characterisation of Igneous Rocks Using Split-Hopkinson Pressure Bar and Digital Image Correlation

**DOI:** 10.3390/ma15228264

**Published:** 2022-11-21

**Authors:** Albin Wessling, Jörgen Kajberg

**Affiliations:** Division of Solid Materials, Luleå University of Technology, 97187 Luleå, Sweden

**Keywords:** rock, dynamic mechanical properties, fracture, high-speed imaging, digital image correlation, Split-Hopkinson pressure bar

## Abstract

The dynamic fracture process of rock materials is of importance for several industrial applications, such as drilling for geothermal installation. Numerical simulation can aid in increasing the understanding about rock fracture; however, it requires precise knowledge about the dynamical mechanical properties alongside information about the initiation and propagation of cracks in the material. This work covers the detailed dynamic mechanical characterisation of two rock materials—Kuru grey granite and Kuru black diorite—using a Split-Hopkinson Pressure Bar complemented with high-speed imaging. The rock materials were characterised using the Brazilian disc and uniaxial compression tests. From the high-speed images, the instant of fracture initiation was estimated for both tests, and a Digital Image Correlation analysis was conducted for the Brazilian disc test. The nearly constant tensile strain in the centre was obtained by selecting a rectangular sensing region, sufficiently large to avoid complicated local strain distributions appearing between grains and at voids. With a significantly high camera frame rate of 671,000 fps, the indirect tensile strain and strain rates on the surface of the disc could be evaluated. Furthermore, the overloading effect in the Brazilian disc test is evaluated using a novel methodology consisting of high-speed images and Digital Image Correlation analysis. From this, the overloading effects were found to be 30 and 23%. The high-speed images of the compression tests indicated fracture initiation at 93 to 95% of the peak dynamic strength for granite and diorite, respectively. However, fracture initiation most likely occurred before this in a non-observed part of the sample. It is concluded that the indirect tensile strain obtained by selecting a proper size of the sensing region combined with the high temporal resolution result in a reliable estimate of crack formation and subsequent propagation.

## 1. Introduction

The process of dynamic fracture of rock materials is important for several industrial applications, e.g., drilling in geothermal and mining industries. For geothermal applications, it is estimated that 50% of the total cost per installed megawatt of energy is associated with drilling and well construction [1], with drill bit wear from hard rock drilling being a predominant cost factor [2]. By utilizing numerical simulations, insights can be gained regarding the hard rock drilling process that otherwise would be difficult to obtain. A recently developed statistical model by Wessling et al. [3], based on the discrete element method, has been shown to capture typical rock fracture behaviour. However, in order for this model to be accurate, precise information regarding the dynamic properties of the rock material is crucial. Specifically, the dynamic compressive and tensile properties alongside information about fracture initiation and propagation are of interest.

Due to the dynamic nature of rock drilling, the experimental characterisation has to be carried out in the dynamic regime. The Split-Hopkinson Pressure Bar (SHPB), or Kolsky bar [4], has historically been used to dynamically characterize various ductile materials at high strain rates (10${}^{1}$–10${}^{4}$ s${}^{-1}$). Today, the SHPB is widely used for dynamic characterisation of rock materials [5]. The International Society for Rock Mechanics and Rock Engineering (ISRM) suggests two tests for characterizing the dynamic tensile and compressive strengths of rock materials—the Brazilian Disc Test (BDT) and the Uniaxial Compression Test (UCT) [6]. Dai et al. [7] evaluated the feasibility of using these tests for dynamic characterisation and concluded that, with proper sample dimensions, pulse shaping techniques and bar interface lubrication, the BDT and UCT constitute a reliable framework for evaluating the dynamic strengths of rock materials. Several recent studies applying the dynamic UCT can be found in the literature, such as the study of the dynamic compressive behavior of basalt rock [8], veined rocks from underground mines [9] and heated sandstone [10]. Recent publications using the dynamic BDT for rock materials can also be found in the literature for studying the fracture process in igneous rocks [11], evaluating the dynamic tensile stress–strain curve of granite [12] and obtaining the dynamic tensile elastic modulus of marble [13]. Compared to direct tensile testing methods, the BDT provides a more practical approach for evaluating the tensile strength of rock materials. However, researchers have found that the indirect tensile strength measured from the BDT overestimates the true tensile strength [14,15]. For dynamic testing, this overestimation has been attributed to the overloading phenomenon, where the disc continues to carry load even after fracture initiation in the disc, after which the well-known Griffith failure criterion is invalid. By identifying the time of crack initiation using strain-gauges, researchers have previously been able to compensate for the overloading effect [16,17].

With the use of high-speed photography, the SHPB can be complemented with information regarding fracture initiation and propagation. High-speed photography was used as early as 1968 by Perkins and Green [18], where it was applied to capture a single picture per dynamic UCT of limestone. More recently, frame rates between 10,000 and 200,000 fps have been used to study the dynamic compressive behavior of rock materials [7,8,9,19,20,21,22,23,24,25]. For the indirect tension, frame rates between 10,000 and 263,000 fps can be found in the literature of dynamic rock testing [7,11,12,19,20,24,26,27,28,29,30,31]. To the best of the present authors’ knowledge, the maximum frame rate used for dynamic rock testing found in literature is the study conducted by Ju et al. [32], where the crack propagation in a notched semi-circular bend specimen was studied using a frame rate of 380,000 fps. Further information, such as exterior full-field measurements regarding deformations of the sample, can be extracted from the high-speed photographs by utilizing Digital Image Correlation (DIC) [33,34]. Digital image correlation has recently been applied to a wide range of dynamic rock experiments [11,17,19,30,31]. Xing et al. [35] combined two high-speed cameras for a 3D-analysis of the full field measurement of the arched surface of a dynamic compression test. Ju et al. [32] used DIC on photographs taken at 381,000 fps, which, as aforementioned, is the highest frame rate the present authors have found in literature.

In this work, an extensive dynamic characterisation of two rock materials—Kuru grey granite and Kuru black diorite—is presented. The objectives are to obtain their mechanical compressive and tensile properties and finally to estimate instants of fracture initiation using a high-speed camera with a significantly high frame rate of 671,000 fps. The compressive properties are obtained directly from the UCT using SHPB, whereas the tensile properties are determined from the BDT. Instead of mounting strain gauges on the flat surface, DIC is used in combination with the high-speed imaging to determine the tensile strain. The tensile strain is obtained according to the recently published novel methodology described by Li et al. [12]. Hence, a rectangular sensing region (RSR) in the centre of the disc is selected, whose size is set sufficiently small to detect the nearly constant strains in the centre, and at the same time large enough to avoid complicated local strain distributions appearing between grains and at voids. However, a wide range of publications using a combination of high-speed photography, DIC and SHPB for rock material characterisation is found in the literature, and the use of a frame rate higher than 381,000 fps [32] remains unexplored. Since rock materials undergo only small deformations before fracture, an increased temporal resolution allows for a more detailed measurement of strain on the surface. An increased temporal resolution also allows for more temporal data points that will increase the accuracy of the determination of fracture initiation and subsequent propagation in the samples. However, as a consequence of the high frame rate, the spatial resolution and the observed region is limited. A set of dynamic BDT at a lower rate of 381,000 fps, capturing the full face of the disc, will therefore be conducted initially to confirm that the limited region is large enough to cover the RSR. By benefitting from the improved precision due to an increased frame rate when detecting initiation of cracks, it will be demonstrated that the so-called overloading effect can be determined using high-speed photography, as opposed to strain gauge methods previously used [16,17]. That is, the overestimation of tensile strength obtained in the standard procedure for BDT [14,15]. Finally, the dynamic increase factors are obtained by comparing the dynamic mechanical properties to the quasi-static counterparts.

## 2. Materials and Methods

In this section, the studied rock materials are described together with the methodology used to investigate the dynamic mechanical properties. This includes relevant theory about the SHPB system and equipment, the UCT, the BDT and the digital image correlation analysis.

### 2.1. Rock Materials and Samples

Two rock materials were investigated in this study—Kuru Gray Granite and Kuru Black Diorite. The granite has an average grain size of 2.0 mm and originates from Niemikylä, Finland, whereas the diorite has an average grain size of 2.4 mm and originates from Ikonen, Finland. The main constituent minerals of the granite and diorite are presented in Table 1 and Table 2, respectively, and the grain textures are shown in Figure 1. All produced rock samples had a diameter of 25 mm, which satisfies the recommendation of having specimen diameters at least 10 times larger than the average rock grain size [6,36,37]. Furthermore, the samples were carefully extracted from larger blocks using a bore machine. Following the recommendation by ISRM [36], ISRM [37], the quasi-static samples were cut into slices of 10.0 mm and 62.5 mm for the BDT and UCT, respectively. The dynamic samples followed the recommendation by Zhou et al. [6], where sample thicknesses of 12.5 mm and 25.0 mm were used for the BDT and UCT, respectively. A grinding machine was used to ensure that the ends of the specimens were smooth before testing. For the dynamic BDT, the samples were sandblasted before applying a black and white DIC speckle pattern.

### 2.2. Quasi-Static Experiments

As a reference to the dynamic testing, the BDT and UCT were also conducted using quasi-static loading. These tests followed the methodologies suggested by ISRM [36], ISRM [37]. For the BDT, an electro-mechanical loading machine (Dartec M1000/RE) with a 22 kN loading cell was used to apply a load at a velocity of 0.2 mm/min. Curved loading jaws with a diameter 2.1 times larger than the sample were utilized in order to distribute the load over a finite arc, which reduces the risk of cracks initiating from stress concentrations at the loading contact [38]. Additionally, a small piece of paper was placed between the sample and jaws, which is known to reduce the stress concentrations further [39]. For the UCT, a compressive loading was applied using a hydraulic machine with a capacity of 4.5 MN combined with a 100 kN load cell. Two opposite circular plates with a spherical seat were used to compress the sample at a velocity of 0.05 mm/min. Rosette strain gauges were mounted on opposite sides of the sample to measure axial and circumferential strains.

### 2.3. Dynamic Testing Using Split-Hopkinson Pressure Bar

The SHPB [4] was used for the dynamic compression and indirect tensile testing. The configuration consists of a 3 m long incident bar and a 2 m long transmitted bar, between which the sample is sandwiched, and a projectile; see Figure 2. The bars as well as the projectile are made of maraging steel with diameters of 25.4 mm. Hence, the length to diameter ratios of the bars are at least 80 and the assumption of one-dimensional wave propagation is valid [40]. The cylindrical projectile of length 150 mm is accelerated by an air gun and impacts the incident bar, generating a compressive stress wave. The wave propagates through the incident bar and is partly reflected and partly transmitted through the sample. By using strain gauges (measurements group CEA-06-250UW-350, 6.35 mm) mounted on the bars, the strain histories of the input $\varepsilon_{i}\left(t\right)$, reflected $\varepsilon_{r}\left(t\right)$ and transmitted $\varepsilon_{t}\left(t\right)$ waves are recorded and amplified (Measurements group 2210A bandwidth 100 kHz). In this study, the signals were sampled with a frequency of 1 MHz. Based on the longitudinal wave speed in the bars and the length of the strain gauges, the time resolution of the wave measurements was approximately 1 μs, which is in line with the sampling rate. The contact interfaces between sample and bars were lubricated in order to reduce the dynamic confinement [7].

For traditional SHPB testing of ductile materials, a rectangular wave is generated in the incident bar. This type of loading is suitable for metals undergoing significant plastic strain before failure. However, for brittle materials such as rock, the steep rise of the rectangular pulse may produce a nonuniform strain rate in the sample and failure may occur before equilibrium is obtained [41]. Dai et al. [7] observed premature failure in the rock sample when a rectangular pulse was used, which demonstrates the importance of proper pulse shaping for dynamic testing of brittle materials. Proper pulse shaping can be achieved by using a cone-shaped projectile [42,43,44] or by placing a thin ductile material at the interface of incident bar impacted by the projectile [45]. The latter was used in this study, i.e., thin aluminum discs were placed between the projectile and incident bar.

In order to obtain the instants of observed crack initiation and full-field deformation measurements, a high-speed camera, Phantom vision v2512, equipped with a microscopic lens, was placed in front of the samples. Two different frame rates, 381,000 and 671,000 fps with resolutions 256 × 128 and 128 × 64 pixels, respectively, were used. In order to obtain sufficient illumination, two LED sources, Constellation 120E15 with 22,000 lm, were directed at the samples. An infrared sensor was used to trigger both the strain data collection and the high-speed camera. The indirect strain in the dynamic BDT was evaluated from a DIC analysis of the high-speed images using the commercial software Aramis (v6.3) by GOM correlate.

### 2.4. Dynamic Uniaxial Compression Test

From elementary wave theory and the definition of one-dimensional strain, the time-shifted forces at the interfaces between sample and bars can be expressed as [40]
(1)$$P_{i}\left(t\right)=AE\left(\varepsilon_{i}\left(t\right)+\varepsilon_{r}\left(t\right)\right)$$
and
(2)$$P_{t}\left(t\right)=AE\varepsilon_{t}\left(t\right)$$
where $P_{i}$ and $P_{t}$ are the forces at the incident and transmitted bar interfaces, respectively. *A* is the cross-sectional area and *E* is the elastic modulus of the bars. Assuming equilibrium, the compressive stress in the dynamic UCT sample is calculated from the strain measured in the transmitted bar as [40]
(3)$$\sigma \left(t\right)=\frac{AE\varepsilon_{t}\left(t\right)}{A_{s}}$$
where $A_{s}$ is the cross-sectional area of the sample. The strain rate is obtained as
(4)$$\dot{\varepsilon }=\frac{2c_{b}\varepsilon_{r}\left(t\right)}{l_{s}}$$
where $c_{b}$ is the longitudinal wave velocity in the bars and $l_{s}$ is the instantaneous sample length. Note that, for the UCT, the convention of compressive strain being positive was used. The axial strain is finally obtained by integrating Equation (Equation 4),
(5)$$\varepsilon =\int_{t_{start}}^{t_{end}}\dot{\varepsilon }\;dt$$

For the UCT, an impact velocity of roughly 23 m/s was used together with a pulse shaper consisting of a 1 mm aluminum shim with an area of 50 mm${}^{2}$, which resulted in a sample loading rate of approximately 11,500 GPas${}^{-1}$. Representative force pulses in the bars using this configuration can be seen in Figure 3a, which originates from a test of diorite. Each test was verified to be in a state of equilibrium up until the sample fracture by comparing the sum of the incident and reflective pulses $P_{i}\left(t\right)+P_{r}\left(t\right)$ with the transmitted pulse $P_{t}\left(t\right)$. An example of this for the diorite can be seen in Figure 3b, where a reasonable equilibrium exists up until a visible fracture (typically around 60 μs).

### 2.5. Dynamic Brazilian Disc Test

For the dynamic Brazilian disc test, the tensile stress at the centre of the disc can be obtained as [6,46]
(6)$$\sigma_{t}=\frac{2P(t)}{\pi Dt}$$

Assuming force equilibrium, P(t) can be evaluated as the averaged bar end loads $\left(P_{1}\left(t\right)+P_{2}\left(t\right)\right)/2$. The sample diameter and thickness are represented by *D* and *t*, respectively. For these tests, an impact velocity of 11 m/s was used together with a pulse shaper consisting of a 0.5 mm aluminum shim with an area of 50 mm${}^{2}$. This resulted in a loading rate of approximately 1450 GPa$s^{-1}$, obtained as the slope of the transmitted wave, see Figure 4b. Representative examples of the pulses obtained in the dynamic BDT are shown in Figure 4b, which originates from a granite test. Each test was verified to be in a state of equilibrium up until sample fracture by comparing the sum of the incident and reflective pulses $P_{i}\left(t\right)+P_{r}\left(t\right)$ with the transmitted pulse $P_{t}\left(t\right)$. An example of this can be seen in Figure 4b, where equilibrium prevails up until a visible fracture (typically around 75 μs).

In addition to the high-speed photography, a subsequent DIC-analysis was conducted on the dynamic BDT with the objective of obtaining the indirect tensile strain. DIC is a non-destructive optical method for full-field exterior measurements of sample deformations. The key aspect of DIC is that there exists a unique and random speckle pattern on the surface which is used to track the displacement between two images. Various speckle sizes have been used in the literature, e.g., 4–5 pixels [12], 5 pixels [47] and 3 pixels [48]. In this study, a stochastic speckle pattern was generated by applying thin layers of black and white spray paint, which resulted in the high contrast speckle seen in Figure 5. By conducting an auto correlation of this speckle-pattern, the average speckle diameter can be estimated to the diameter of the peak at a correlation value of 0.5, see Figure 6a [49]. This resulted in speckle sizes around 3 to 4 pixels for both resolutions used in this study, i.e., 256 × 128 pixels for 381,000 fps and 128 × 64 pixels for 671,000 fps. The DIC-analysis was conducted using the commercial software Aramis (v6.3) by GOM.

By following the method recently proposed by Li et al. [12], the indirect tensile strain was measured as an average of several Virtual Strain Gauges (VSG) with lengths $h_{c}$ distributed over a width $w_{c}$, see Figure 5. Hence, a so-called rectangular sensing region (RSR) was defined, large enough to avoid local effects, such as voids and microcracks or singular grains, but small enough to capture the nearly constant indirect tensile strain region close to the centre of the disc. Furthermore, it is crucial that the region covers the point of crack initiation, which does not always coincide with the centre of the disc [3,15]. To this end, five tests of each rock were conducted with the high-speed camera set to 381,000 fps in order to capture the full surface of the disc, see Figure 5. The horizontal distance *w* between the disc centre and the point of observed crack initiation, see Figure 7a, was determined for each of these tests and are presented in Table 3 and Table 4 for the granite and diorite, respectively. Based on this, the critical width $w_{c}$ for the granite was chosen as 2×1.62=3.24 mm and 2×1.72=3.44 mm for the diorite. The effect of the lengths of the VSG:s was also studied. An example of this can be seen in Figure 7b, where the indirect tensile strain versus gauge length is presented at different times during the test. If the length of the gauges is too small, the measurement is sensitive to the local effects which manifests itself in the fluctuations of the strain measurement. Due to the decrease in indirect strain away from the centre, the measured strain decreases as the length of the VSG:s increases. The optimal length, i.e., that gives little fluctuations and captures the indirect strain at the centre, of each of the tests are presented in Table 3 and Table 4 for the granite and diorite, respectively. Based on this, the critical gauge lengths was chosen as 5.73 mm for the granite and 5.19 mm for the diorite. It should be noted that the aforementioned sizes of the RSR:s can be captured with the FOV of the camera set to 671,000 fps, see Figure 5. Therefore, a frame rate of 671,000 fps was used for the remainder of the indirect tensile testing in this study.

## 3. Results and Discussion

In this section, the results from the study are presented. This includes high-speed images and mechanical properties obtained from the dynamic Brazilian disc tests and uniaxial compression tests.

### 3.1. Brazilian Disc Test

With the frame rate of the camera set to 671,000 fps, a total of six and five Brazilian disc tests were conducted for the granite and diorite, respectively. For each one of these tests, force equilibrium was ensured. Furthermore, the points of observed crack initiation for these tests did not deviate from the RSR defined in Section 2.5, see Figure 5. A representative example of the time evolution of the indirect tensile stress from Equation (Equation 6) and the strain as obtained from the DIC-analysis from one of the Brazilian disc tests of granite can be seen in Figure 8. In this particular example, a crack could be observed from the images at 71.5 μs or at 93.6% of the peak stress, see Figure 8b. However, this is a subjective identification and the crack could have initiated from non-observable parts of the sample. The fully propagated crack can be seen in Figure 8c.

By analysing the strain measurement in Figure 8, it is clear that there are two distinct regions with different strain rate measurements of 65.8 and 1462 s${}^{-1}$. An explanation for this is that the crack actually initiates at the knee of the strain curve and that the measured strain after this point in time time might not be valid. Instead, the strain measurement after this point is a measurement of the relative displacement between two semi-discs. It is well-known that the dynamic BDT can significantly overpredict the tensile strength due to an overloading effect, where the disc continues to carry load even after fracture initiation [17]. If, for the example in Figure 8, the crack initiates at the knee of the strain measurement curve, the fracture initiation occurs at 71.6% of the peak stress value, which is in agreement with previous studies [17]. Because of this, the strain measure in this work will only be considered valid up until the instant of probable crack initiation.

The full-field strain evolution on the face of the dynamic BDT of granite (same sample as in Figure 8) can be seen in Figure 9a,b, and the corresponding tensile stress–strain curve is presented in Figure 9i. From Figure 9a–d, it can be seen that two points of maximum tensile strain are obtained close to the bar interfaces, which is consistent with what has been observed in numerical simulations of the strain distribution before crack initiation [15]. After this, up until the image with the first observed crack in Figure 10b, the strain concentrates to the centre part of the disc. However, as mentioned in the above discussion, the strain measurement after the instant of probable crack initiation is dubious.

The stress–strain curves from all of the dynamic BDT are presented in Figure 10, and a summary of the results can be found in Table 5. Taking into account the heterogeneous nature of the materials, the results are fairly consistent with regard to the initial linear part of the curve, probable crack initiation, observed crack initiation and peak stress level. On average, the peak stresses were measured to be 38.1 and 39.8 MPa for granite and diorite, respectively. However, by taking the overloading effect into account, using the instant of probable crack initiation, the actual average tensile strengths were found to be 26.6 and 30.5 MPa, corresponding to overloading effects of 30 and 23%. Here, the overloading effect is defined as the ratio of the overload to the peak tensile strength. Using the strain measure up until the instant of probable crack initiation, the indirect tensile strain rates were measured to be 115 and 51 s${}^{-1}$ for granite and diorite, respectively. Compared to the quasi-static peak strengths of 16.9 and 22.8 MPa, these strain rates resulted in the dynamic increase factors of 1.57 (granite) and 1.34 (diorite).

### 3.2. Dynamic Uniaxial Compression

In total, seven and ten dynamic compression tests were conducted for the granite and diorite, respectively. All of the tests were ensured to be in equilibrium. A representative example of the dynamic compressive testing of diorite can be seen in Figure 11. The high-speed images (381,000 fps) of the fracture process can be seen in Figure 11a–h and the corresponding stress–strain and strain rate curves are presented in Figure 11i. The first observed crack could be observed at 59.75 μs at a stress level of 277 MPa or 93.5 % of the peak value (red square in Figure 11i). However, it is important to note that, since this is a subjective identification and only one arched side of the sample surface was observed with the camera, this observation is not an absolute determination of the instant of fracture initiation. It is likely that the fracture initiated before this elsewhere, i.e., at a non-observed part of the sample. A more precise estimate of the instant of fracture initiation could probably be obtained by analysing strain data as obtained from a DIC-analysis, similarly to what was achieved for the BDT. However, since the sample surface is arched, this would require two synchronous high-speed cameras and a stereo analysis of the results [35]. After the observed fracture initiation, the crack can be seen to propagate in Figure 11c–h alongside the initiation and propagation of new cracks. This multiple crack growth is well known for dynamic compressive testing of heterogeneous brittle materials and is attributed to the inertia effect associated with crack tip opening [41]. Figure 11i shows that the strain rate is nearly constant during the instant of fracture initiation; hence, the sample was deforming uniformly during the testing.

The stress–strain curves and the instants of observed fracture initiation for all of the dynamic compression tests are presented in Figure 12, and a summary of the results is presented in Table 6. At the instants of fracture initiation, the strain rates were found to be on average 260 s${}^{-1}$ and 282 s${}^{-1}$ for the granite and diorite, respectively, and the dynamic compressive peak strengths were found to be 343 and 313 MPa, respectively.This dynamic compressive strength of Kuru granite is consistent with previous studies on the same material [50] For the granite, the quasi-static peak strength was found to be 216 MPa, resulting in a dynamic increase factor of 1.48, while the quasi-static strength of the diorite was 164 MPa, resulting in a dynamic increase factor of 1.91. Hence, the dynamic peak strength of the diorite is more sensitive to strain rate compared to that of granite. On average, the first crack could be observed at 320 and 297 MPa, corresponding to 93 and 95% of the peak dynamic strength, for the granite and diorite, respectively. Although, as mentioned above, the fracture most likely initiated earlier.

## 4. Conclusions

The objective of this study was to obtain the dynamic characterisation of two rock materials—Kuru grey granite and Kuru black diorite. This was achieved by utilizing the uniaxial compression and Brazilian disc test in a combined SHPB, high-speed photography and DIC configuration.

The recently published method [12] to quantify the tensile strains in the BDT:S provided reliable values at a very high temporal resolution (671,000 fps). Furthermore, this enhanced temporal resolution allowed for a sufficiently large data set to obtain the indirect tensile strain and strain rate before fracture from the DIC-analysis;A novel method using high-speed photography and DIC to evaluate the overloading effect in BDT:s was demonstrated. This resulted in overloading effects of 30 and 23% for the granite and diorite, respectively;From the high-speed images of the compression tests, the fracture initiations were detected at 90 to 95% of the peak strength levels. However, it is likely that the fracture initiation occurred earlier at some non-observed part of the sample.

## Figures and Tables

**Figure 1 materials-15-08264-f001:**
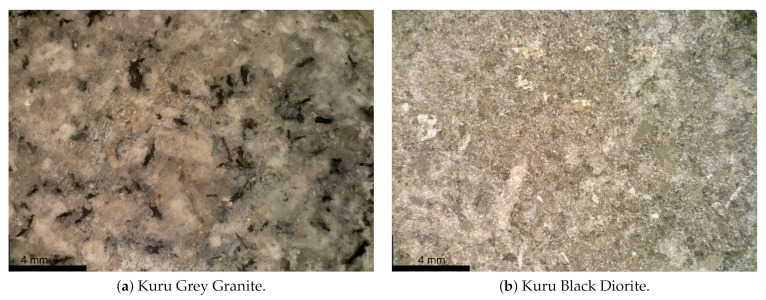
The grain textures of the rocks characterised in this study.

**Figure 2 materials-15-08264-f002:**
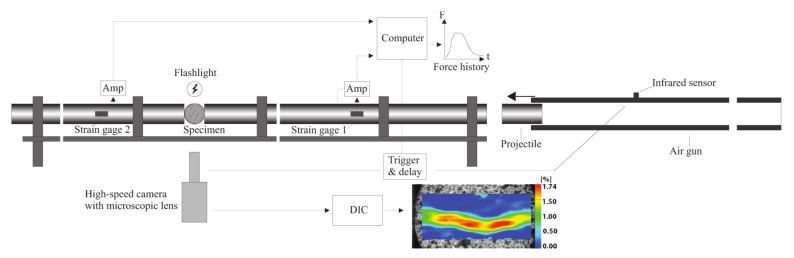
The Split-Hopkinson pressure bar configuration used for the dynamic testing. Here, the set-up is shown for the Brazilian disc test. The setup is essentially the same for the uniaxial compression test but without DIC processing of the images.

**Figure 3 materials-15-08264-f003:**
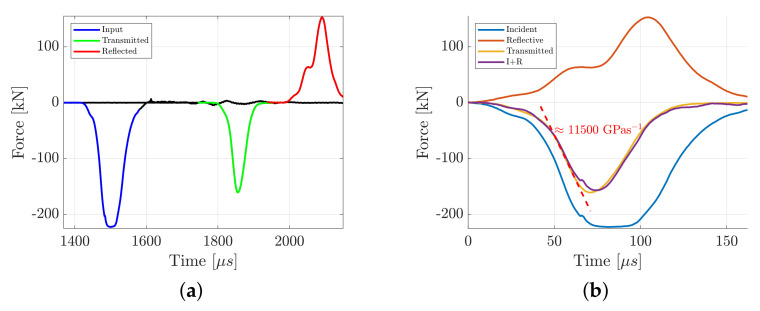
Example pulses from the dynamic uniaxial compression test of diorite, which is representative for all the compressive testing conducted in this study. The incident, transmitted and reflective force waves (**a**) and the validation of force equilibrium in the sample by comparing the pulses (**b**).

**Figure 4 materials-15-08264-f004:**
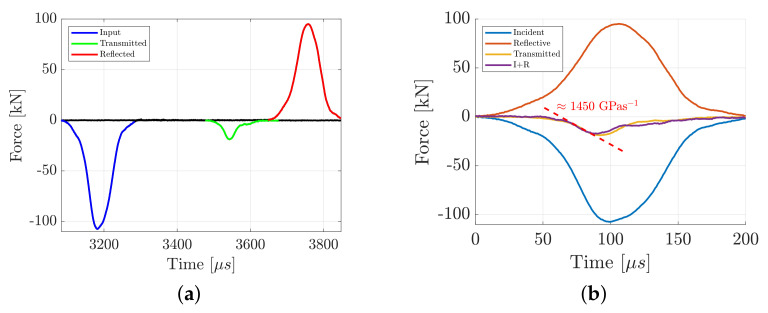
Representative pulses from the dynamic Brazilian disc test of granite, which is representative for all the tests conducted in this study. The incident, transmitted and reflective force waves (**a**) and the validation of force equilibrium in the sample by comparing the pulses (**b**).

**Figure 5 materials-15-08264-f005:**
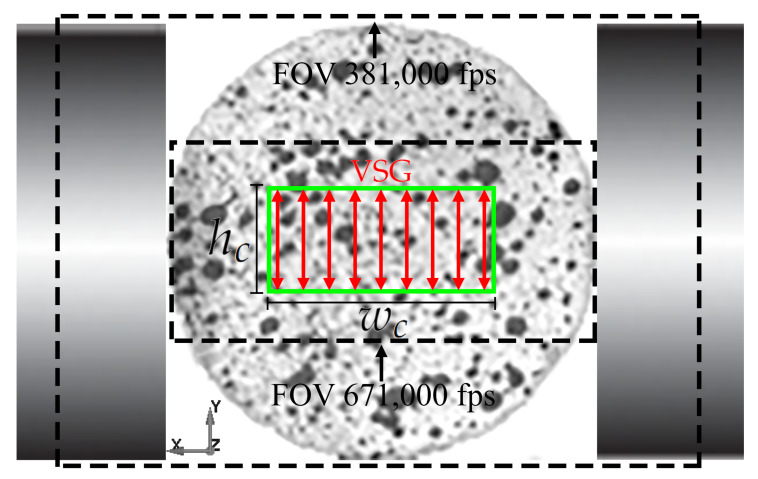
An example of a speckle on the dynamic Brazilian disc tests, the two Field of Views (FOV) used for the high-speed photography and the RSR (green rectangle).

**Figure 6 materials-15-08264-f006:**
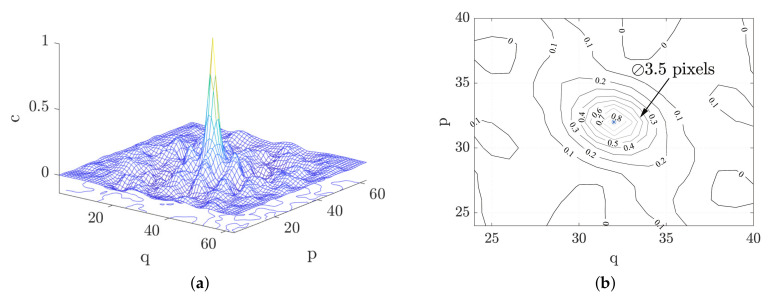
A typical correlation peak obtained from the speckle patterns used in this study (**a**) and the correlation peak projected onto a plane (**b**).

**Figure 7 materials-15-08264-f007:**
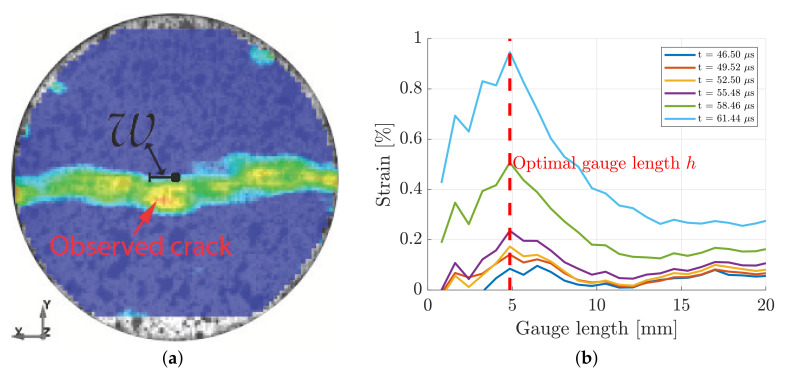
An example of the measurement of the horizontal distance from the centre to the point of observed crack initiation in the Brazilian disc test of granite (**a**) and an example of the effects of gauge length for measuring the indirect strain of granite (**b**).

**Figure 8 materials-15-08264-f008:**
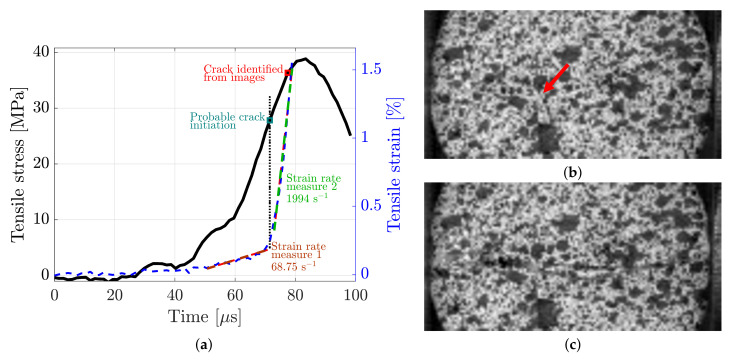
An example of the time evolution of indirect tensile stress and strain from a Brazilian disc test of granite (**a**), the instant of observed crack initiation (**b**) and the fully propagated crack (**c**). The resolution of the high-speed images (**b**,**c**) is 128 × 64 pixels or 26.0 × 13.0 mm.

**Figure 9 materials-15-08264-f009:**
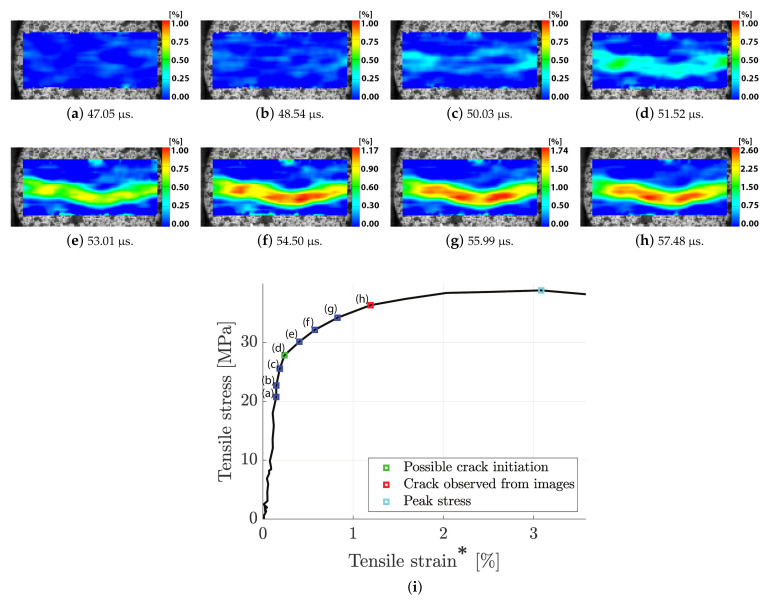
An example of high-speed photography and DIC of a Brazilian disc test of granite (**a**–**h**) together with the corresponding indirect tensile stress versus tensile strain (**i**). The ${}^{*}$ denotes that the use of the term strain after crack initiation is dubious since the measurement is only valid up until the instant of crack initiation. Note that the use of the term strain after crack initiation is dubious since the measurement is only valid up until the instant of initiation. The resolution of the high-speed images (**a**–**h**) is 128 × 64 pixels or 26.0 × 13.0 mm.

**Figure 10 materials-15-08264-f010:**
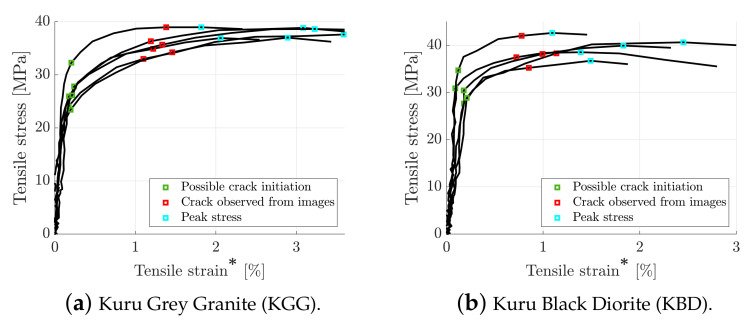
Dynamic indirect tensile stress versus tensile strain for the investigated rocks. The ${}^{*}$ denotes that the use of the term strain after crack initiation is dubious since the measurement is only valid up until the instant of initiation.

**Figure 11 materials-15-08264-f011:**
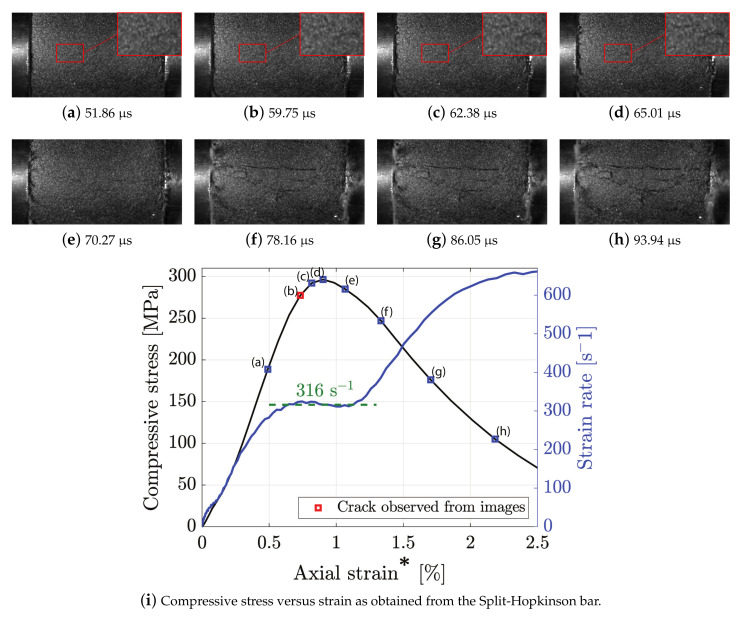
Examples of high-speed images of a dynamic compression test of diorite (**a**–**h**) and the corresponding stress–strain curve (**i**). The resolution of the high-speed images (**a**–**h**) is 256 × 128 pixels or 31.5 × 15.8 mm.

**Figure 12 materials-15-08264-f012:**
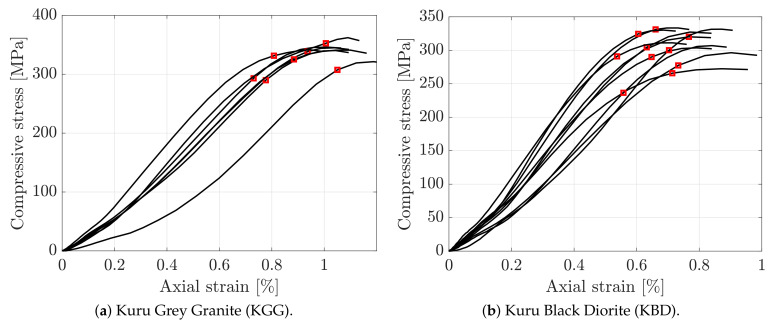
Dynamic compressive stress versus axial strain for the investigated rocks. The red star marks the instant at which crack initiation could be observed in the high-speed images.

**Table 1 materials-15-08264-t001:** Mineral composition of Kuru Grey Granite (KGG).

Mineral	Plagioclase	Quartz	Potassium Feldspar	Biotite	Muscovite	Other
wt %	32.6	31.0	30.8	3.5	1.2	0.9

**Table 2 materials-15-08264-t002:** Mineral composition of Kuru Black Diorite (KBD).

Mineral	Plagioclase	Biotite	Amphibians	Ilmenite	Climopyroxene	Other
wt %	57.0	21.0	10.0	6.4	6.0	4.0

**Table 3 materials-15-08264-t003:** The distance between the point of observed crack initiation and disc centre and optimal gauge lengths for the granite. The critical values used to define the RSR are marked in bold font.

Granite Sample	1	2	3	4	5
*w* [mm]	1.60	**1.62**	0.41	0.42	0.80
*h* [mm]	5.66	**5.73**	5.70	4.89	4.86

**Table 4 materials-15-08264-t004:** The distance between the point of observed crack initiation and disc centre and optimal gauge lengths for the diorite. The critical values used to define the RSR are marked in bold font.

Diorite Sample	1	2	3	4	5
*w* [mm]	0.86	0.43	0.87	1.30	**1.72**
*h* [mm]	5.15	**5.19**	3.45	4.31	5.18

**Table 5 materials-15-08264-t005:** Summary of quasi-static and dynamic indirect tensile tests.

	Quasi-Static	Dynamic			
**Rock**	**Tensile** **Strength** **[MPa]**	**Probable** **Crack Initiation** **[MPa]**	**Observed Crack** **Initiation Stress** **[MPa]**	**Peak** **Stress** **[MPa]**	**Tensile** **Strain Rate** **[s** ${}^{-1}$ **]**
Granite	16.9	26.6 ± 3.2	35.5 ± 2.0	38.1 ± 0.9	115 ± 34
Diorite	22.8	30.5 ± 2.7	38.2 ± 2.5	39.8 ± 2.1	51 ± 13

**Table 6 materials-15-08264-t006:** Summary of quasi-static and dynamic uniaxial compression tests.

	Quasi-Static			Dynamic		
**Rock**	**Peak** **Strength** **[MPa]**	**Elastic** **Modulus** **[GPa]**	**Poisson’s** **Ratio** **[-]**	**Observed Fracture** **Initiation Stress** **[MPa]**	**Peak** **Strength** **[MPa]**	**Compressive** **Strain Rate** **[s** ${}^{-1}$ **]**
Granite	216	63.1	0.29	320 ± 24	343 ± 12	260 ± 26
Diorite	164	79.4	0.28	297 ± 31	313 ± 19	282 ± 32

## Data Availability

Not applicable.

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
