# Peer review of "Dynamic Compressive and Tensile Characterisation of Igneous Rocks Using Split-Hopkinson Pressure Bar and Digital Image Correlation"

_materials, 2022, doi:10.3390/ma15228264_

Round 1

Reviewer 1 Report

The research "Dynamic Compressive and Tensile Characterisation of Igneous Rocks using Split-Hopkinson and Digital Image Correlation" is meaningful, but the research scheme design in this paper is relatively rough, and some contents need to be supplemented.

1. In this paper, the research status of Igneous Rocks using Split-Hopkinson and Digital Image Correlation should be highlighted to illustrate the necessity of this paper in the Introduction.

2. The abstract should be rewritten to include the problem definition, methodology, main findings and significant results. I leave my suggestion to the authors for reflection.

3. Figure 8, 9, 10. Please, add the dimension marker. 

Author Response

Response to Reviewer 1 Comments
Thank you for taking the time to assess our manuscript. Your feedback has greatly helped us to improve the text. Listed below are the comments from your review along with our responses. Furthermore, we have uploaded a latex-file with track changes on for your convinience.

Point 1: In this paper, the research status of Igneous Rocks using Split-Hopkinson and Digital Image Correlation should be highlighted to illustrate the necessity of this paper in the Introduction
Response 1: We agree that the research status should have been highlighted more. In the revised version you will find we extended parts of the introduction in order to rectify this.

Point 2: The abstract should be rewritten to include the problem definition, methodology, main findings and significant results. I leave my suggestion to the authors for reflection.
Response 2: We agree that the problem definition, methodology, findings and significant results were not clearly stated in the abstract. In the revised version, we have made the abstract more concise in order to emphasize your suggestions. 

Point 3:  Figure 8, 9, 10. Please, add the dimension marker. 
Response 3: We assume that you mean Figure 11 instead of Figure 10 since Figure 10 already has dimension markers. We have added dimension markers for Figure 8, 9 and 11. 

Reviewer 2 Report

This manuscript mainly studied the dynamic mechanical properties alongside information about the initiation and propagation of cracks of Kuru Grey Granite and Kuru Black Diorite using the Split-Hopkinson Pressure Bar complemented with high-speed imaging. The instants of fracture initiation are estimated and the dynamic increase factors are obtained by comparing the dynamic mechanical properties to the quasi-static counterparts. This manuscript is interesting and is well written. I think it can be published in present form.

Author Response

Response to Reviewer 2 Comments
Thank you for taking the time to assess our manuscript and your positive feedback

Reviewer 3 Report

In my opinion, the manuscript can be considered for publication as the content grabs the reader’s attention. However, some issues need to be addressed by authors prior to acceptance for publication:

1. The abstract is quite lengthy, can be more concise in  directly addressing the essence of the study. In addition, please add short conclusion.

2. The problem statement and novelty of the work is not obvious in the manuscript. What is the difference between this study and previous studies using dynamic regime Split-Hopkinson. Can be highlighted further.

3. It is suggested to include relevant figure, photo or diagram to support the introduction.

4. Methodology is clear and provides the whole idea about procedures.

5. Overall the discussion is interesting, however in can be improved a bit by adding more in-depth discussion to clarify and justify the results, especially, on dynamic uniaxial compression.

6. The conclusion can be written more concise. The conclusion only need to state and answer the objective without the deep and details. Some parts in the current conclusions can be removed. In addition, the conclusion can include the following, the new concepts and innovations from this study as part of the  concluding remark.

7. References are acceptable and up-to-date.

8. The length of paragraphs which should not be too short or too long.

Author Response

Response to Reviewer 3 Comments
Thank you for taking the time to assess our manuscript. Your feedback has greatly helped us to improve the text. Listed below are the comments from your review along with our responses. Furthermore, we have uploaded a latex-file with track changes on for your convinience.

Point 1: The abstract is quite lengthy, can be more concise in  directly addressing the essence of the study. In addition, please add short conclusion.
Response 1: We agree that the abstract was lengthy and that it affected the emphasis on the essence of the study. Following your suggestion, we have made the abstract more concise and emphasised the essence of the study as well as added a short conclusion.

Point 2: The problem statement and novelty of the work is not obvious in the manuscript. What is the difference between this study and previous studies using dynamic regime Split-Hopkinson. Can be highlighted further.
Response 2: We agree that the differences from previous studies should have been highlighted better. In the revised version we have rewritten the introduction in order to rectify this. Specifically, we emphasized the need for our camera setup for rock materials and related it to what other researchers have used. We also emphasized the novelty of our method for obtaining the overloading effect.

Point 3: It is suggested to include relevant figure, photo or diagram to support the introduction.
Response 3:
Thank you for your suggestion. We believe that images and tables are better suited in the Materials and methods section.

Point 4: Methodology is clear and provides the whole idea about procedures.
Response 4: 
Thank you for your feedback.

Point 5: Overall the discussion is interesting, however in can be improved a bit by adding more in-depth discussion to clarify and justify the results, especially, on dynamic uniaxial compression.
Response 5:
We have included more discussion regarding the compression tests. Specifically, we discuss how the experimental setup could possibly be improved in order to obtain a more accurate estimation of the fracture initiation. We have also compared the measured strength values of the granite rock with results from previous published papers.

Point 6: The conclusion can be written more concise. The conclusion only need to state and answer the objective without the deep and details. Some parts in the current conclusions can be removed. In addition, the conclusion can include the following, the new concepts and innovations from this study as part of the  concluding remark.
Response 6:
We agree that the conclusions should have been more concise. We have rewritten the whole conclusions section and made it more concise. We have also emphasized the new concepts and innovations.

Point 7: References are acceptable and up-to-date.
Response 7:
Thank you for your feedback.

Point 8: The length of paragraphs which should not be too short or too long.
Response 8:
We noticed that some of the paragraphs were a bit too short and we have rectified this in the revised version.